# A decline in tuberculosis diagnosis, treatment initiation and success during the COVID-19 pandemic, using routine health data in Cape Town, South Africa

Karen Jennings[1,2], Martina Lembani[2], Anneke C. Hesseling[3], Nyameka Mbula[4], Erika Mohr-Holland[5], Vanessa Mudaly[4], Mariette Smith[4,6], Muhammad Osman[3,7], Sue-Ann Meehan[3]*

1 City of Cape Town Health Department, Specialised Health, HIV/STI/TB Unit, Cape Town, South Africa, 2 School of Public Health, University of Western Cape, Cape Town, South Africa, 3 Department of Paediatrics and Child Health, Desmond Tutu TB Centre, Faculty of Medicine and Health Sciences, Stellenbosch University, Cape Town, South Africa, 4 Department of Health and Wellness, Provincial Government of the Western Cape, Cape Town, South Africa, 5 City of Cape Town Health Department, Specialised Health, Epidemiology Unit, Cape Town, South Africa, 6 Department of Public Health and Facility Medicine, University of Cape Town, Cape Town, South Africa, 7 School of Human Sciences, Faculty of Education, Health and Human Sciences, University of Greenwich, London, United Kingdom

* sueannm@sun.ac.za

**Data Availability Statement:** These data were provided for analysis by the Western Cape Department of Health, Provincial Health Data

## Abstract

### Background

Coronavirus disease (COVID-19) negatively impacted tuberculosis (TB) programs which were already struggling to meet End-TB targets globally. We aimed to quantify and compare diagnosis, treatment initiation, treatment success, and losses along this TB care cascade for drug-susceptible TB in Cape Town, South Africa, prior to and during COVID-19.

### Methods

This observational study used routine TB data within two predefined cohorts: pre-COVID-19 (1 October 2018–30 September 2019) and during-COVID-19 (1 April 2020–31 March 2021). The numbers of people diagnosed, treated for TB and successfully treated were received from the Western Cape Provincial Health Data Centre. Pre and post treatment loss to follow up and cascade success rates (proportion of individuals diagnosed with an outcome of treatment success) were calculated and compared across cohorts, disaggregated by sex, age, HIV status, TB treatment history and mode of diagnosis.

### Results

There were 27,481 and 19,800 individuals diagnosed with drug-susceptible TB in the pre- and during-COVID-19 cohorts respectively, a relative reduction of 28% (95% CI [27.4% - 28.5%]). Initial loss to follow up increased from 13.4% to 15.2% (p<0.001), while post treatment loss increased from 25.2% to 26.1% (p < 0.033). The overall cascade success rate

Centre. Whilst the data are anonymised, they are highly granular health data linked to individual health care clients in the Province and, crucially, no informed consent has been given for research use of these routine health data. For this reason the Western Cape Department of Health does not permit open sharing but instead only grants primary use permission for the data (no permission was granted for secondary use). The data are too sensitive to be openly shared. Other users would need to make the application themselves and each requested use case is reviewed by the Department of Health. Re-use of this dataset requires approval from the PHDC (Provincial Health Data Centre), and Dr Moodley, Director: HIA, Western Cape Department of Health, South Africa can be contacted to advise on this process (email: melvin.moodley@westerncape. gov.za).

**Funding:** This publication was supported by the South African Department of Science and Innovation (DSI) and the South African Medical Research Council (SAMRC) under the BRICS JAF #2020/101. Prof. Anneke C. Hesseling received the award. The content and findings reported herein are the sole deduction, view and responsibility of the researcher/s and do not reflect the official position and sentiments of the funders.https:// www.samrc.ac.za/.

**Competing interests:** The authors have declared that no competing interests exist.

dropped by 2.1%, from 64.8% to 62.7% (p< 0.001). Pre- and during-COVID-19 cascade success rates were negatively associated with living with HIV and having recurrent TB.

## Conclusions

An already poorly performing TB program in Cape Town was negatively impacted by the COVID-19 pandemic. There was a substantial reduction in the number of individuals diagnosed with drug-susceptible. Increases in pre-and post-treatment losses resulted in a decline in TB cascade success rates. Strengthened implementation of TB recovery plans is vital, as health services now face an even greater gap between achievements and targets and will need to become more resilient to possible future public health disruptions.

## Introduction

The COVID-19 pandemic became a public health crisis in 2020, with the SARS-CoV-2 virus accounting for nearly 7 million deaths worldwide by mid 2023 [1], making it the leading cause of death from a single infectious agent in 2020 and 2021, eclipsing TB, which had been the leading infectious agent since 2011 [2]. Global reductions in TB mortality and morbidity in the decade 2010–19 were insufficient to be on track to reach ambitious "End-TB strategy" and Sustainable Development Goal (SDG) targets [3], which has been exacerbated by COVID-19, with fewer people being tested, diagnosed and treated [4]. In countries supported by the Global Fund to fight AIDS, Tuberculosis, and Malaria, there was a decline from 2019 to 2020 in key TB program indicators, which was unprecedented since the inauguration of the fund in 2002 [5]. Globally, case notifications declined by 18%; from 7.1 million in 2019 to 5.8 million in 2020, a 9-year setback in plans as outlined by the Stop TB partnership to End TB [6, 7]. There have been some gains in finding and treating people with TB (PWTB), showing some degree of return to pre-COVID-19 figures in 2021 and a more substantial recovery in 2022; the number of people treated for TB had increased to 6.4 million in 2021 and 7.5 million in 2022 [8].

South Africa is one of the top ten high-burden countries for TB, HIV-associated TB and drug-resistant TB [6]. Under a South African National State of Disaster, various levels of lockdown were legislated from 26 March 2020 to 5 April 2022, resulting in limited access to public health services and negative impacts on health programs, including TB testing and diagnosis [9]. Initial COVID-19 restrictions resulted in a ∼48% average weekly decrease in TB Xpert MTB/RIF testing volumes while the number of TB positive tests declined by 33% [10]. Early evidence suggested that the disruption of routine TB services would lead to suboptimal retention in care and poorer treatment outcomes for individuals with TB in South Africa [11], and globally [12].

South African provincial level data, while limited, showed considerable declines in TB testing and treatment patterns with much heterogeneity across provinces [13]. Using epidemiological data from a local context can inform and help prioritize key aspects within TB programs which require improvement [14]. The aim of this study was to quantify and compare the number of individuals diagnosed with drug- susceptible tuberculosis (DS-TB), treated for TB, successfully completed treatment; and the losses in the continuum of TB care in a large metropolitan district in South Africa (Cape Town), prior to and during the COVID-19 pandemic.

## Methods

### Study design

This was an observational study comparing two retrospective annual DS-TB cohorts; pre-COVID-19 (1 October 2018 to 30 September 2019) and during-COVID-19 (1 April 2020 to 31 March 2021) cohorts. Pre-COVID-19 reflected the most recent period possible while including sufficient time to record the 6-month TB treatment outcomes. The during-COVID-19 period was pragmatically defined to start from 1 April 2020, following the onset of the first lockdown on 26 March 2020 [15]. Using the TB care cascade as a conceptual framework [16–18], the study quantified and compared the number of people diagnosed with TB; treated for TB; and with treatment success across cohorts. We also calculated the treatment success rate and the cascade success rate for each annual period. In addition, we analysed the losses between diagnosis and treated for TB (initial loss to follow up, ILTFU) and between treated for TB and treatment success (post-treatment loss, PTL) comparing cohorts, and disaggregated by sex, age, HIV status, previous TB history and mode of diagnosis.

### Setting

Cape Town is a metropolitan district within the Western Cape (WC) Province of South Africa. It is home to 4.7 million people (66% of the provincial population) [19]. Primary health care services are rendered by two health authorities, namely the Western Cape Government Health and Wellness Department (WCGHW) and the City of Cape Town (CCT) municipal government [20]. In the WC, 90% of the population access TB testing in the public sector services [20].

In 2019, there were 29,408 DS and 1,509 drug resistant (DR) TB diagnoses [21] in Cape Town. DS-TB treatment success was 76.3% in the 2018 cohort [22]. Antenatal HIV prevalence was 22% [23] with 54% ART coverage [20].

Unique to the WC, the Department of Health houses a Provincial Health Data Centre (PHDC), which links various primary sources of data (including laboratory, pharmacy, hospital and primary healthcare information and electronic disease-specific registers), using a unique patient identifier, into single patient level records [24], thus providing a more complete, integrated, person-level data set than any individual data system.

### Data collection

Data was obtained from the PHDC (11 October 2022) for all individuals diagnosed with DS-TB in two predefined cohorts: the first was pre-COVID-19 and the second was during-COVID-19. All data received was de-identified; authors could not identify individual participants.

### Variables, definitions

Study variables and definitions are provided in Table 1.

### Data analysis

The PHDC undertook de-duplication of the data, using the unique patient identifier. Potential data anomalies were checked against source data. The numbers of individuals who were diagnosed with DS-TB, treated for TB and with treatment success within each cohort were quantified. Further, ILTFU and PTL were calculated and compared across the cohorts with Chi Squared tests using the IBM SPSS 29 software package. The open-source web tool OpenEpi was used to calculate the 95% confidence limits for proportions.

**Table 1. Definitions of key variables.**

| Key term | Definition |
|---|---|
| Diagnosed with TB | The number of individuals diagnosed with drug-susceptible TB (DS-TB), including with bacteriologically confirmed and clinically diagnosed TB, pulmonary and extra-pulmonary TB, new and retreatment TB and those diagnosed at hospital and primary care level. |
| Treated for TB | The number of individuals diagnosed with DS-TB for whom there was evidence of TB treatment initiation at a site of TB registration and notification (i.e. a primary health care facility or TB-hospital, which in South Africa have been the TB registration and notification sites and custodians of TB treatment registers). |
| Treatment success | The number of individuals with microbiological evidence of TB cure or who were recorded as TB treatment completion at 6 months. |
| Treatment success rate | The proportion of individuals who were treated for TB during the study period who had a documented outcome of treatment success. |
| Cascade success rate | The proportion of individuals diagnosed with TB during the study period who had a documented outcome of treatment success. |
| Initial loss to follow-up (ILTFU) | Individuals diagnosed with TB for whom there was no evidence of having started TB treatment at a site of TB treatment. |
| Post-treatment loss (PTL) | Individuals with TB for whom there was evidence of TB treatment initiation but without TB treatment success. |

TB, Tuberculosis.

## Ethics

Ethics approval for this study was obtained from the University of the Western Cape (UWC) Biomedical Research Ethics Committee (BM21/10/19). Study approval was also received from the Western Cape Provincial Health and City Health Departments (WC_202107_037; 9453). A waiver of informed consent was obtained for the routine health data received from the PHDC. All data received from the PHDC was anonymised.

## Results

There were 27,481 individuals diagnosed with DS-TB in the pre-COVID-19 cohort and 19,800 individuals in the during-COVID-19 cohort. Demographics and disease characteristics were similar for individuals diagnosed with TB in both cohorts; men accounted for more than 50%, children <15 years accounted for 10% of the cohort, 40% of individuals were living with HIV, 75% were new TB diagnoses, and 70% were bacteriologically confirmed diagnoses (Table 2).

### Pre-COVID-19 period

ILTFU was 13.4% and PTL was 25.2%. The collective losses resulted in a treatment success rate of 74.8% and a cascade success rate of 64.8% (Table 3).

ILTFU was significantly higher among children (<15 years) compared to adults (≥15 years), while PTL was significantly higher among adults compared to children. Both ILTFU and PTL were significantly higher among individuals living with HIV compared to HIV-negative individuals, individuals with TB retreatment. compared to those with new TB and those with a bacteriological diagnosis compared to those with a clinical diagnosis (Table 4). Overall cascade success was significantly higher among HIV-negative people compared to those living with HIV, those with new TB compared to those with retreatment and those who were clinically diagnosed compared to those who were bacteriologically diagnosed (Table 4).

**Table 2. Demographic and clinical characteristics of individuals diagnosed with DS-TB stratified by pre- and during-COVID-19 periods, in Cape Town, South Africa.**

| | | COVID-19 PERIOD | | | |
| | | Pre-COVID -19 (1 October 2018 to 30 September 2019 | | During-COVID-19 (1 April 2020 to 31 March 2021) | |
| | | (n) | (%) | (n) | (%) |
|---|---|---|---|---|---|
| Total number of individuals diagnosed with DS-TB | | 27 481 | 100.0% | 19 800 | 100% |
| **Sex*** | Female | 11 849 | 43.1% | 8 445 | 42.7% |
| | Male | 15 581 | 56.7% | 11 310 | 57.1% |
| **Age*** | Child (<15) | 2 589 | 9.4% | 1 663 | 8.4% |
| | Adult (≥15) | 24 891 | 90.6% | 18 131 | 91.6% |
| **HIV status** | Negative | 11 286 | 41.1% | 8 678 | 43.8% |
| | Positive | 11 781 | 42.9% | 8 509 | 43.0% |
| | Unknown | 4 414 | 16.1% | 2 613 | 13.2% |
| **Previous TB history** | New | 21 220 | 77.2% | 14 946 | 75.5% |
| | Retreatment | 6 261 | 22.8% | 4 854 | 24.5% |
| **Mode of diagnosis** | Bacteriological | 19 541 | 71.1% | 14 503 | 73.2% |
| | Clinical | 7 940 | 28.9% | 5 297 | 26.8% |

* very small amount of missing data.

COVID-19, Coronavirus disease of 2019; DS-TB, Drug- susceptible tuberculosis; HIV, Human immunodeficiency virus.

## Comparing the pre-COVID-19 and during-COVID-19 periods

S1 and S2 Tables show the during-COVID-19 period data (as Tables 3 and 4 do for the pre-COVID-19 period). Fig 1 shows a relative reduction of 28% (95% CI [27.4% - 28.5%]) in the number of individuals diagnosed with DS-TB from pre-COVID-19 to during-COVID-19.

Overall, ILTFU increased from 13.4% pre-COVID-19 to 15.2% during-COVID-19 ($p < 0.001$), a relative increase of 13.2%. ILTFU increased significantly irrespective of sex (males p<0.001, females p = 0.002), age (child p = 0.007, adult p<0.001) and HIV status (HIV-positive p<0.001, HIV-negative p = 0.032). The relative increase in ILTFU was higher for males (14.6%) compared to females (11.4%) and for those living with HIV (22.2%) compared

**Table 3. DS-TB diagnosed, treated, and treatment success during the pre-COVID-19 period (October 2018 to September 2019), in Cape Town, South Africa, disaggregated by demographic and clinical characteristics.**

| Variable | | Diagnosed with DS-TB | ILTFU n (%) | Treated for TB | PTL n (%) | Treatment success n (%) | Cascade success |
|---|---|---|---|---|---|---|---|
| **total** | | **27 481** | **3 683 (13.4%)** | **23 798** | **5 989 (25.2%)** | **17 809 (74.8%)** | **64.8%** |
| **Sex** | Female | 11 849 | 1 648 (13.9%) | 10 201 | 2 484 (24.4%) | 7 717 (75.6%) | 65.1% |
| | Male | 15 581 | 2 032 (13.0%) | 13 549 | 3 494 (25.8%) | 10 055 (74.2%) | 64.5% |
| **Age, years** | Child (<15) | 2 589 | 491 (19.0%) | 2 098 | 369 (17.6%) | 1 729 (82.4%) | 66.8% |
| | Adult (≥15) | 24 891 | 3 192 (12.8%) | 21 699 | 5 619 (25.9%) | 16 080 (74.1%) | 64.6% |
| **HIV status** | HIV negative | 11 286 | 820 (7.3%) | 10 466 | 2 329 (22.3%) | 8 137 (77.7%) | 72.1% |
| | HIV positive | 11 781 | 1 692 (14.4%) | 10 089 | 2 916 (28.9%) | 7 173 (71.1%) | 60.9% |
| **Previous TB history** | New | 21 220 | 2 773 (13.1%) | 18 447 | 4 463 (24.2%) | 13 984 (75.8%) | 65.9% |
| | Retreatment | 6 261 | 910 (14.5%) | 5 351 | 1 526 (28.5%) | 3 825 (71.5%) | 61.1% |
| **Mode of Diagnosis** | Bacteriological | 19 541 | 2 728 (14.0%) | 16 813 | 4 434 (26.4%) | 12 379 (73.6%) | 63.3% |
| | Clinical | 7 940 | 955 (12.0%) | 6 985 | 1 555 (22.3%) | 5 430 (77.7%) | 68.4% |

COVID-19, Coronavirus disease of 2019; DS-TB, Drug- susceptible tuberculosis; HIV, Human immunodeficiency virus; ILTFU, Initial loss to follow-up; PTL, Post-treatment loss.

**Table 4. Comparing initial loss to follow up (ILTFU), post-treatment loss (PTL) and cascade success for all individuals diagnosed with DS-TB during the pre-COVID-19 period in Cape Town, South Africa, disaggregated by demographic and clinical characteristics.**

| Variable | | Initial loss to follow up (ILTFU) | p value* | Post-treatment loss (PTL) | p value* | Cascade success | p value* |
|---|---|---|---|---|---|---|---|
| **Total** | | 13.4% | | 25.2% | | 64.8% | |
| **Sex** | Female | 13.9% | **0.037** | 24.4% | **0.012** | 65.1% | 0.307 |
| | Male | 13.0% | | 25.8% | | 64.5% | |
| **Age** | Child (<15) | 19.0% | **<0.001** | 17.6% | **<0.001** | 66.8% | **0.027** |
| | Adult (≥15) | 12.8% | | 25.9% | | 64.6% | |
| **HIV status** | HIV negative | 7.3% | **<0.001** | 22.3% | **<0.001** | 72.1% | **<0.001** |
| | HIV positive | 14.4% | | 28.9% | | 60.9% | |
| **Previous TB history** | New | 13.1% | **0.003** | 24.2% | **<0.001** | 65.9% | **<0.001** |
| | Retreatment | 14.5% | | 28.5% | | 61.1% | |
| **Mode of diagnosis** | Bacteriological | 14.0% | **<0.001** | 26.4% | **<0.001** | 63.3% | **<0.001** |
| | Clinical | 12.0% | | 22.3% | | 68.4% | |

\* p < 0.05 in bold font.

COVID-19, Coronavirus disease of 2019; DS-TB, Drug- susceptible tuberculosis; HIV, Human immunodeficiency virus; ILTFU, Initial loss to follow-up; PTL, Post-treatment loss; TB, Tuberculosis.

to HIV-negative individuals (11.2%). ILTFU was significantly higher among new TB patients compared to retreatment (p<0.001) and those who were clinically diagnosed compared to those bacteriologically diagnosed (p<0.001) (Table 5).

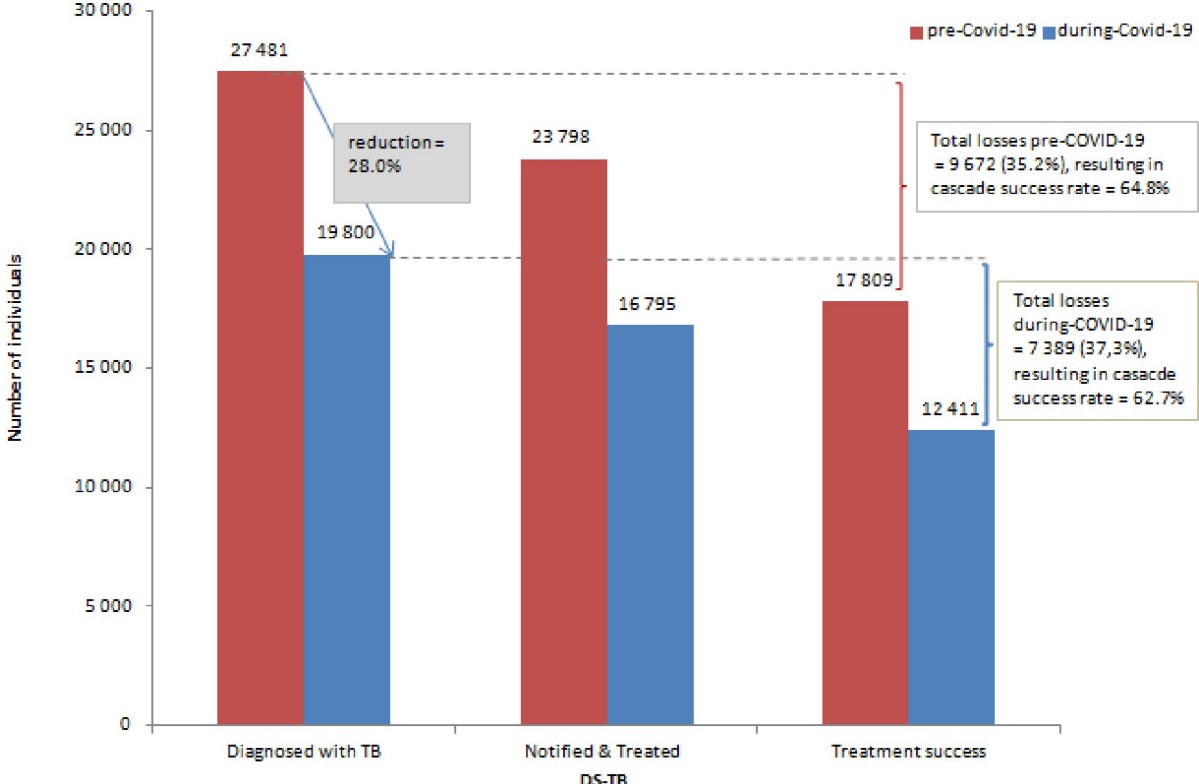

**Fig 1. Pre- & during-COVID-19 individuals diagnosed with DS-TB, treated for TB and treatment success across Cape Town, South Africa.** COVID-19, Coronavirus disease of 2019; DS-TB, Drug- susceptible tuberculosis.

**Table 5. Increases in initial loss to follow up (ILTFU) and post-treatment loss (PTL) between pre- & during-COVID-19 periods, disaggregated by demographic and clinical characteristics for all individuals diagnosed with DS-TB in Cape Town, South Africa.**

| Variable | | Losses | Pre- COVID-19 | During- COVID-19 | p value* | Absolute Increase | Relative Increase |
|---|---|---|---|---|---|---|---|
| **Total** | | ILTFU | 13.4% | 15.2% | **<0.001** | 1.8% | 13.2% |
| | | PTL | 25.2% | 26.1% | **0.033** | 0.9% | 3.7% |
| **Sex** | Male | ILTFU | 13.0% | 14.9% | **<0.001** | 1.9% | 14.6% |
| | Female | | 13.9% | 15.5% | **0.002** | 1.6% | 11.4% |
| | Male | PTL | 25.8% | 26.7% | 0.122 | 0.9% | 3.5% |
| | Female | | 24.4% | 25.3% | 0.150 | 1.0% | 3.9% |
| **Age** | Child (<15) | ILTFU | 19.0% | 22.4% | **0.007** | 3.4% | 18.0% |
| | Adult(≥15) | | 12.8% | 14.5% | **<0.001** | 1.7% | 13.2% |
| | Child (<15) | PTL | 17.6% | 19.4% | 0.194 | 1.8% | 10.1% |
| | Adult (≥15) | | 25.9% | 26.7% | 0.098 | 0.8% | 3.0% |
| **HIV status** | HIV positive | ILTFU | 14.4% | 17.5% | **<0.001** | 3.2% | 22.2% |
| | HIV negative | | 7.3% | 8.1% | **0.032** | 0.8% | 11.2% |
| | HIV positive | PTL | 28.9% | 29.8% | 0.203 | 0.9% | 3.1% |
| | HIV negative | | 22.3% | 23.7% | **0.017** | 1.5% | 6.7% |
| **Category TB** | Retreatment | ILTFU | 14.5% | 15.2% | 0.340 | 0.6% | 4.5% |
| | New | | 13.1% | 15.2% | **<0.001** | 2.1% | 16.1% |
| | Retreatment | PTL | 28.5% | 29.8% | 0.189 | 1.2% | 4.3% |
| | New | | 24.2% | 24.9% | 0.145 | 0.7% | 3.0% |
| **Mode diagnosis** | Bacteriological | ILTFU | 14.0% | 14.7% | 0.066 | 0.7% | 5.1% |
| | Clinical | | 12.0% | 16.6% | **<0.001** | 4.5% | 37.8% |
| | Bacteriological | PTL | 26.4% | 27.0% | 0.265 | 0.6% | 2.2% |
| | Clinical | | 22.3% | 23.7% | 0.072 | 1.5% | 6.5% |

* p values < 0.05 in bold font.

HIV, Human immunodeficiency virus; ILTFU, Initial loss to follow-up PTL, Post-treatment loss; TB, Tuberculosis.

PTL increased from 25.2% pre-COVID-19 to 26.1% during-COVID-19 (p = 0.033), a relative increase of 3.7%. PTL in HIV negative individuals increased from 22.3% pre-COVID-19 to 23.7% during-COVID-19 (p = 0.017), a relative increase of 6.7% (Table 5).

The treatment success rate decreased from 74.8% to 73.9%, a relative reduction of 1.3%. The cascade success rate dropped from 64.8% pre-COVID-19 to 62.7% during COVID-19 (p< 0.001), a relative reduction of 3.3% (Fig 1).

## Discussion

COVID-19 has had a substantial epidemiological impact on the TB program in Cape Town, a high TB burden district in South Africa. Comparing pre-COVID-19 to during-COVID-19, the number of individuals diagnosed with TB decreased by 28%. ILTFU increased by 13.2% and PTL by 3.7% during COVID-19. Treatment success decreased by 1.3% and cascade success by 3.3%.

Understanding the impact of COVID-19 at a sub-national level is vital to strengthen the district level TB program and allows for a focused implementation of the South African TB recovery plan [25, 26]. This will also provide opportunities to continue addressing gaps in the care cascade, which existed pre-COVID-19, and to address the losses experienced during COVID-19. Access to consolidated TB-related data through the PHDC in this setting provided the unique opportunity to comprehensively understand the changes in some of the steps in the TB care cascade in Cape Town, making this study practically relevant for health professionals,

policy makers and civil society in this context, but also relevant TB programs in similar high TB-burdened settings.

When considering the 28% decline in the number of people diagnosed with DS-TB during-COVID-19, it is important to note that the WC Province reported a 31% decrease in the number of individuals accessing primary health care facilities in between March and December 2020 compared to the same period in the previous year [9], which led to a decreased opportunity for programmatic TB screening and testing and thereby TB diagnosis, a pattern which was also seen nationally in South Africa [27] and globally [28]. Other studies from the Cape Metro also looked into the trends over this period for RR-TB and found similar findings [29]. Limitations of reduced population mobility and reduced access to health services because of lockdown measures, the prioritization of COVID-19 services away from other health services including TB, the prioritization of 'urgent' care, as well as changes in health seeking behaviour [9, 11, 30, 31] are plausible reasons that could explain this drop.

Prior to COVID-19, TB incidence was declining and COVID-19 negatively impacted on this trend [8]. South Africa's response to this global concern includes a focused TB recovery plan [25] as well as the incorporation of key TB recovery strategies into the National Strategic Plan for HIV, TB and STIs (2023–2028) [32], for example, accelerating the implementation of targeted universal TB testing (TUTT), which tests high risk individuals for TB (living with HIV, close TB contacts and previous TB), irrespective of TB symptoms [33].

In this study, ILTFU increased significantly from 13.4% pre-COVID-19 to 15.2% during-COVID-19. This proportion of ILTFU is similar to the 12% estimated nationally for individuals with DS-TB pre-COVID [18], but lower than a recent study conducted in two health sub-districts of Cape Town (2018–2020) which found ILTFU to be 20% among those diagnosed with DS- and RR-TB [34]. We found that the relative increase in ILTFU was higher among males, and individuals living with HIV. Health seeking behaviour is typically poorer among males compared to females [35, 36] we know that a lower proportion of males compared to females seek TB care in South Africa [37]. It is plausible that COVID-19 exacerbated this problem.

Reasons for delayed linkage to care include lack of information and support from health care providers, unpleasant previous TB treatment episodes and being uncertain of their TB diagnosis [38]. Mortality is also a driver of ILTFU, as individuals who are sicker may die before they can link to care. Increasing age, HIV positive status, and a hospital-based TB diagnosis have been identified as predictors of mortality [39]. Males and individuals living with HIV have also been identified as sub-groups having a higher risk of mortality due to COVID-19 itself [40]. Strategies to find men specifically, early diagnosis and strengthening linkage of people diagnosed with TB to treatment, including the strengthening referral pathways from hospitals to mitigate ILTFU, are part of the TB recovery plan for South Africa [41] and are vital TB recovery strategies in South Africa.

PTL increased by 3.7% and the COVID-19 lockdown factors that impacted access to healthcare generally could have played a role in the reduced number of individuals completing their TB treatment. This reduction is relatively small, which may be explained by some of the interventions put in place to mitigate the impact of COVID-19-related factors and retain individuals in care, including policies to keep PHC services functional for essential services such as HIV and TB, in addition to the already established practice in Cape Town of monthly dispensing, following a 2 week treatment initiation phase, instead of daily directly observed therapy [42].

The proportion of individuals started on TB treatment with a successful TB treatment outcome (treatment success rate) decreased by 1.3%, from a below par success rate pre-COVID-19. TB treatment success is routinely reported for all individuals who are notified and started on treatment; however this study has additionally calculated a cascade success rate out of the

diagnosed cohort. We found that the cascade success rate significantly declined by 3.3% during-COVID-19. Less than two thirds of those diagnosed with TB successfully completed treatment. Recording and reporting successful TB treatment outcomes as a proportion of those diagnosed, not only those initiated on treatment, can focus attention on addressing ILTFU [34] as well as the more routinely recognised PTL.

One of the main strengths of the study was the use of PHDC data (instead of data from stand-alone routine TB programmatic reporting systems). This allowed the inclusion of TB diagnoses made on clinical grounds and obviated modelling. Using PHDC data allowed for a cohort-based approach, with the same individuals followed from diagnosis to outcome, as has been recommended in constructing TB care cascades, to minimise risk of bias [16]. An additional strength of the study was that the data was extracted more than a year after the cohort periods, obviating the potential problem of missing data relating to reporting lags.

This study has limitations. We did not estimate TB incidence rates (i.e. the number of people diagnosed with TB in an annual period per 100 000 population). Population size decreases (e.g. due to COVID-19 mortality and out-migration) could theoretically have contributed to the reduction which was observed in the number of people diagnosed with TB between the two study-periods, however any population decrease is unlikely to have offset the trend of estimated annual population year-on-year increases [43] and highly unlikely to be on the same scale as the 28% decline in the number of people diagnosed with DS-TB in the annual period following the onset of the COVID-19 pandemic. Another limitation is that this study used data from only the public sector. While the vast majority of TB diagnosis and treatment is in the public sector [18], possible changes in sector use during the study period may be a confounder which could not be ascertained. Missing data can also be a limitation when using routine health data.

## Conclusion

This study provides important findings on the impact of COVID-19 on TB services in a metropolitan district of South Africa. We found that COVID-19 negatively impacted TB health services, reducing diagnoses, and increasing pre- and post-treatment losses, resulting in a decline in individuals successfully being treated for TB.

During COVID-19, there was a substantial drop in the number of people diagnosed with DS-TB and an increase in losses along the DS-TB continuum of care, which was more marked for ILTFU than for PTL. This differential loss along the TB care cascade may reflect programmatic prioritisation of retaining people in care rather than focussing on linking newly diagnosed people to care. This highlights an important opportunity for mitigating risk in future service interruptions and the importance of TB programs designing interventions to ensure continuity of care across the cascade.

Within the research community, it remains important to determine what underlies the decrease in people diagnosed with TB. Within TB programmes, the priority is to ensure undiagnosed TB does not increase to prevent additional morbidity, mortality and onward transmission in communities. Focused implementation and attainment of local and national TB recovery plans are therefore of vital importance.

## Supporting information

**S1 Table. DS-TB diagnosed, treated, and treatment success in the during-COVID-19 period (April 2020 to March 2021), in Cape Town, South Africa, disaggregated by demographic and clinical characteristics.** COVID-19, Coronavirus disease of 2019; DS-TB, Drug-susceptible tuberculosis; HIV, Human immunodeficiency virus; ILTFU, Initial loss to follow-

up; PTL, Post-treatment loss.
(DOCX)

**S2 Table. Comparing initial loss to follow up (ILTFU), post-treatment loss (PTL) and cascade success for all individuals diagnosed with DS-TB in the during-COVID-19 period (April 2020 to March 2021) in Cape Town, South Africa, disaggregated by demographic and clinical characteristics.** * p < 0.05 in bold font; COVID-19, Coronavirus disease of 2019; DS-TB, Drug- susceptible tuberculosis; HIV, Human immunodeficiency virus; ILTFU, Initial loss to follow-up; PTL, Post-treatment loss; TB, Tuberculosis.
(DOCX)

## Acknowledgments

The work of the PHDC staff in making an integrated WCGH data system a reality is acknowledged, and particularly the role of Ms Mariette Smith, who extracted the data from the PHDC for this study. The study was conducted for the lead author's MPH mini-thesis, however it was nested within a larger study entitled the "Epidemiological impact and intersection of the COVID-19 and tuberculosis pandemics in Brazil, Russia, India and South Africa (IMPAC$_{19}$T$_B$)" and support from the larger study team is acknowledged in terms of conceptualisation of the study, ethical approval and application for access to PHDC data, as well as assistance with data validation. Prior and ongoing work in the Provincial and City of Cape Town Departments of Health using data to address gaps in the TB programme is acknowledged.

## Author Contributions

**Conceptualization:** Karen Jennings, Martina Lembani, Muhammad Osman, Sue-Ann Meehan.

**Data curation:** Karen Jennings, Mariette Smith.

**Formal analysis:** Karen Jennings.

**Funding acquisition:** Anneke C. Hesseling.

**Methodology:** Karen Jennings.

**Supervision:** Martina Lembani, Sue-Ann Meehan.

**Validation:** Karen Jennings, Mariette Smith, Muhammad Osman, Sue-Ann Meehan.

**Visualization:** Karen Jennings, Martina Lembani, Nyameka Mbula, Erika Mohr-Holland, Vanessa Mudaly, Mariette Smith, Muhammad Osman, Sue-Ann Meehan.

**Writing – original draft:** Karen Jennings, Muhammad Osman, Sue-Ann Meehan.

**Writing – review & editing:** Karen Jennings, Martina Lembani, Anneke C. Hesseling, Nyameka Mbula, Erika Mohr-Holland, Vanessa Mudaly, Mariette Smith, Sue-Ann Meehan.

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
