## [Decision Letter · Decision Letter 0]

12 Jul 2024

PONE-D-24-18062A decline in tuberculosis diagnosis, treatment initiation and success during the COVID-19 pandemic, using routine health data in Cape Town, South AfricaPLOS ONE

Dear Dr. Meehan,

Thank you for submitting your manuscript to PLOS ONE. After careful consideration, we feel that it has merit but does not fully meet PLOS ONE’s publication criteria as it currently stands. Therefore, we invite you to submit a revised version of the manuscript that addresses the points raised during the review process. Please submit your revised manuscript by Aug 26 2024 11:59PM. If you will need more time than this to complete your revisions, please reply to this message or contact the journal office at plosone@plos.org. Please include the following items when submitting your revised manuscript:A rebuttal letter that responds to each point raised by the academic editor and reviewer(s). You should upload this letter as a separate file labeled 'Response to Reviewers'.A marked-up copy of your manuscript that highlights changes made to the original version. You should upload this as a separate file labeled 'Revised Manuscript with Track Changes'.An unmarked version of your revised paper without tracked changes. You should upload this as a separate file labeled 'Manuscript'.

We look forward to receiving your revised manuscript.

Kind regards,

Veranyuy Ngah, MSc

Academic Editor

PLOS ONE

 [This publication was supported by the South African Department of Science and Innovation (DSI) and the South African Medical Research Council (SAMRC) under the BRICS JAF #2020/101. Prof. Anneke C. Hesseling received the award. The content and findings reported herein are the sole deduction, view and responsibility of the researcher/s and do not reflect the official position and sentiments of the funders.https://www.samrc.ac.za/].  

3. In the online submission form you indicate that your data is not available for proprietary reasons and have provided a contact point for accessing this data. Please note that your current contact point is a co-author on this manuscript. According to our Data Policy, the contact point must not be an author on the manuscript and must be an institutional contact, ideally not an individual. Please revise your data statement to a non-author institutional point of contact, such as a data access or ethics committee, and send this to us via return email. Please also include contact information for the third party organization, and please include the full citation of where the data can be found.

4. We notice that your supplementary figures are uploaded with the file type 'Figure'. Please amend the file type to 'Supporting Information'. Please ensure that each Supporting Information file has a legend listed in the manuscript after the references list.

Additional Editor Comments (if provided):

Reviewers' comments:

Reviewer's Responses to Questions

**Comments to the Author**

1. Is the manuscript technically sound, and do the data support the conclusions?

Reviewer #1: Partly

2. Has the statistical analysis been performed appropriately and rigorously? 

Reviewer #1: Yes

3. Have the authors made all data underlying the findings in their manuscript fully available?

Reviewer #1: No

4. Is the manuscript presented in an intelligible fashion and written in standard English?

Reviewer #1: Yes

5. Review Comments to the Author

Reviewer #1: Thank you for the opportunity to review this topic of importance especially in Cape Town where tuberculosis is very prevalence.

There is a lag period of 5 months between the pre and during covid (October 2019 to March 2020). Five months is almost half the time period of study used by the authors and lot of changes can take place. The authors could provide a brief explanation as to why this time period was omitted.

Methods

Lines 125 to 127 are part of the results and should not appear under methods.

The definition of terms provided is a necessary inclusion.

Results

Figure 1 should be included in the body of the manuscript as this is major results that answers the research question of comparing the two periods

There are no table of results provided showing during Covid-19 period like table 3 and 4 for pre-covid 19. If there are too many then appendix should be provided.

Lines 239-241: The authors are describing results that have not been presented, as there are no results for the covid-19 period.

6. PLOS authors have the option to publish the peer review history of their article (what does this mean?). If published, this will include your full peer review and any attached files.

Reviewer #1: No

---

## [Author Response · Author response to Decision Letter 0]

6 Aug 2024

Our response to reviewers has been uploaded as a separate letter

---

## [Editor Report · Decision Letter 1]

29 Aug 2024

A decline in tuberculosis diagnosis, treatment initiation and success during the COVID-19 pandemic, using routine health data in Cape Town, South Africa

PONE-D-24-18062R1

Dear Dr. Sue-Ann Meehan

We’re pleased to inform you that your manuscript has been judged scientifically suitable for publication and will be formally accepted for publication once it meets all outstanding technical requirements.

Kind regards,

Veranyuy Ngah, MSc

Academic Editor

PLOS ONE
---

## [Editor Report · Acceptance letter]

2 Sep 2024

PONE-D-24-18062R1 

PLOS ONE

Dear Dr. Meehan, 

I'm pleased to inform you that your manuscript has been deemed suitable for publication in PLOS ONE. Congratulations! Your manuscript is now being handed over to our production team.

Kind regards, 

on behalf of

Dr. Veranyuy Ngah 

Academic Editor

PLOS ONE